# Baseline oxygen consumption decreases with cortical depth

**Philipp Mächler** [1,⊕,¤a]**, Natalie Fomin-Thunemann**[1,⊕]**, Martin Thunemann**[1]**, Marte Julie Sætra**[2]**, Michèle Desjardins**[3]**, Kıvılcım Kılıç**[1]**, Layth N. Amra**[1]**, Emily A. Martin**[1]**, Ichun Anderson Chen**[1]**, Ikbal Şencan-Eğilmez**[4]**, Baoqiang Li**[4,¤b]**, Payam Saisan**[5]**, John X. Jiang**[1]**, Qun Cheng**[5]**, Kimberly L. Weldy**[5]**, David A. Boas**[1]**, Richard B. Buxton**[6]**, Gaute T. Einevoll**[7,8]**, Anders M. Dale**[5,6]**, Sava Sakadžić**[4]\***, Anna Devor**[1,4]\*

1 Department of Biomedical Engineering, Boston University, Boston, Massachusetts, United States of America, 2 Department of Numerical Analysis and Scientific Computing, Simula Research Laboratory, Oslo, Norway, 3 Département de Physique, de Génie Physique et d'Optique and Axe Oncologie, Centre de Recherche du CHU de Québec–Université Laval, Université Laval, Québec, Canada, 4 Athinoula A. Martinos Center for Biomedical Imaging, Department of Radiology, Harvard Medical School, Massachusetts General Hospital, Charlestown, Massachusetts, United States of America, 5 Department of Neurosciences, University of California San Diego, La Jolla, California, United States of America, 6 Department of Radiology, University of California San Diego, La Jolla, California, United States of America, 7 Department of Physics, University of Oslo, Oslo, Norway, 8 Department of Physics, Norwegian University of Life Sciences, Ås, Norway

⊕ These authors contributed equally to this work.
¤a Current address: Department of Physics, University of California San Diego, La Jolla, California, United States of America
¤b Current address: Brain Cognition and Brain Disease Institute, Shenzhen Institute of Advanced Technology, Chinese Academy of Sciences, Shenzhen, Guangdong, China
* sava.sakadzic@mgh.harvard.edu (SS); adevor@bu.edu (AD)

**Data Availability Statement:** All data is available on Brain Imaging Library; DOI is 10.35077/g.312 (https://doi.org/10.35077/g.312).

**Funding:** This work was supported by the National Institutes of Health (BRAIN Initiative

## Abstract

The cerebral cortex is organized in cortical layers that differ in their cellular density, composition, and wiring. Cortical laminar architecture is also readily revealed by staining for cytochrome oxidase—the last enzyme in the respiratory electron transport chain located in the inner mitochondrial membrane. It has been hypothesized that a high-density band of cytochrome oxidase in cortical layer IV reflects higher oxygen consumption under baseline (unstimulated) conditions. Here, we tested the above hypothesis using direct measurements of the partial pressure of $O_2$ ($pO_2$) in cortical tissue by means of 2-photon phosphorescence lifetime microscopy (2PLM). We revisited our previously developed method for extraction of the cerebral metabolic rate of $O_2$ ($CMRO_2$) based on 2-photon $pO_2$ measurements around diving arterioles and applied this method to estimate baseline $CMRO_2$ in awake mice across cortical layers. To our surprise, our results revealed a decrease in baseline $CMRO_2$ from layer I to layer IV. This decrease of $CMRO_2$ with cortical depth was paralleled by an increase in tissue oxygenation. Higher baseline oxygenation and cytochrome density in layer IV may serve as an $O_2$ reserve during surges of neuronal activity or certain metabolically active brain states rather than reflecting baseline energy needs. Our study provides to our knowledge the first quantification of microscopically resolved $CMRO_2$ across cortical layers as a step towards better understanding of brain energy metabolism.

R01MH111359 to AD; BRAIN Initiative
U19NS123717 to AD; R01DA050159 to AD;
R01NS108472 to DAB, K99MH120053 to IS;
U24EB028941 to SS, R01NS091230 to SS,
U01HL133362 to SS, R01NS115401 to SS,
RF1NS121095 to SS, R00MH120053 to IS,
R35NS097265 to PM) and Swiss National Science
Foundation (P2ZHP3_181568 to PM). The funders
had no role in study design, data collection and
analysis, decision to publish, or preparation of the
manuscript.

**Competing interests:** The authors have declared
that no competing interests exist.

**Abbreviations:** 2PLM, 2-photon phosphorescence
lifetime microscopy; CBF, cerebral blood flow;
$CMRO_2$, cerebral metabolic rate of $O_2$; COX,
cytochrome oxidase; FITC, fluorescein
isothiocyanate; NA, numerical aperture; $pO_2$, partial
pressure of $O_2$; ROI, region of interest; SBR, signal-
to-background ratio; SI, primary somatosensory
cortex; SR101, sulforhodamine 101.

## Introduction

Neuronal circuits in cerebral cortex are organized in layers that differ in their cellular density, composition, and wiring [1,2], as well as the density of mitochondrial cytochrome oxidase (COX), a marker of $O_2$ metabolism [3]. In primary cortices, the highest density of COX is found in layer IV, and the lowest in layer I [4,5]. Therefore, it is commonly believed that layer IV has higher cerebral metabolic rate of $O_2$ ($CMRO_2$) compared to other layers. These differences have been specifically hypothesized to reflect laminar variation in metabolic costs under baseline (unstimulated) conditions [4]. Experimentally addressing this hypothesis is important for the basic understanding of cerebrocortical energetics.

Experimental measurement of layer-specific (i.e., laminar) $CMRO_2$ has been challenging, because in common practice it requires information about both blood oxygenation and blood flow [6–9]. Previously, we introduced a method for extraction of $CMRO_2$ [10] based on a single imaging modality: 2-photon phosphorescence lifetime microscopy (2PLM) [10–12] providing measurements of the partial pressure of $O_2$ ($pO_2$). This method relies on the Krogh–Erlang model of $O_2$ diffusion from a cylinder [13], which assumes that a volume of tissue within a certain radius around a blood vessel gets all its $O_2$ from that blood vessel. Previously, we applied our method to estimate $CMRO_2$ in the rat cerebral cortex, where capillaries were absent within an approximately 100-µm radius around penetrating arterioles, satisfying the model assumption. The laminar profile of $CMRO_2$, however, was not addressed due to limited penetration of our imaging technology at that time.

In the present study, we increased our penetration depth by (i) utilizing a new $pO_2$ probe Oxyphor 2P with red-shifted absorption and emission spectra [14], (ii) switching from rats to mice because mice have a thinner cortex and smaller diameter of surface blood vessels attenuating light, and (iii) optimizing our procedure of probe delivery to avoid spilling of the probe on the cortical surface, which exacerbates out-of-focus excitation. With these improvements, we performed $pO_2$ measurements in fully awake mice in cortical layers I–IV.

In the mouse cortex, periarteriolar spaces void of capillaries are narrower than those in the rat [15,16]. This challenges the basic Krogh–Erlang assumption of the center arteriole serving as the sole $O_2$ source. Therefore, we revisited the model to account for the contribution of the capillary bed and then applied the revised model to quantify the baseline laminar $CMRO_2$ profile. Our results show that, contrary to the common notion [4], baseline $CMRO_2$ in layer IV does not exceed that in the upper cortical layers. We speculate that higher COX density in layer IV may reflect higher $O_2$ metabolism during transient surges of neuronal activity or certain metabolically active brain states rather than baseline energy needs.

## Results

### Depth-resolved measurements of tissue $pO_2$ in awake mice with chronic optical windows

We used 2PLM (Fig 1A) [10–12] in combination with a new $pO_2$ probe, Oxyphor 2P [14], to image baseline interstitial (tissue) $pO_2$ in the primary somatosensory cortex (SI) around diving arterioles at different depths, from the cortical surface down to approximately 500 µm deep. All measurements were performed in awake mice with chronically implanted optical windows (Figs 1B, 1C and S1). Oxyphor 2P was pressure-microinjected into the cortical tissue approximately 400 µm below the surface and was allowed to diffuse, resulting in sufficient signal intensity in cortical layers I–IV within approximately 40 min after the injection. Sulforhodamine 101 (SR101) was co-injected with Oxyphor 2P to label astrocytes as a means for real-time monitoring of tissue integrity. In addition, fluorescein isothiocyanate (FITC)–labeled dextran was injected intravenously to visualize the vasculature. We imaged $pO_2$ using either square or

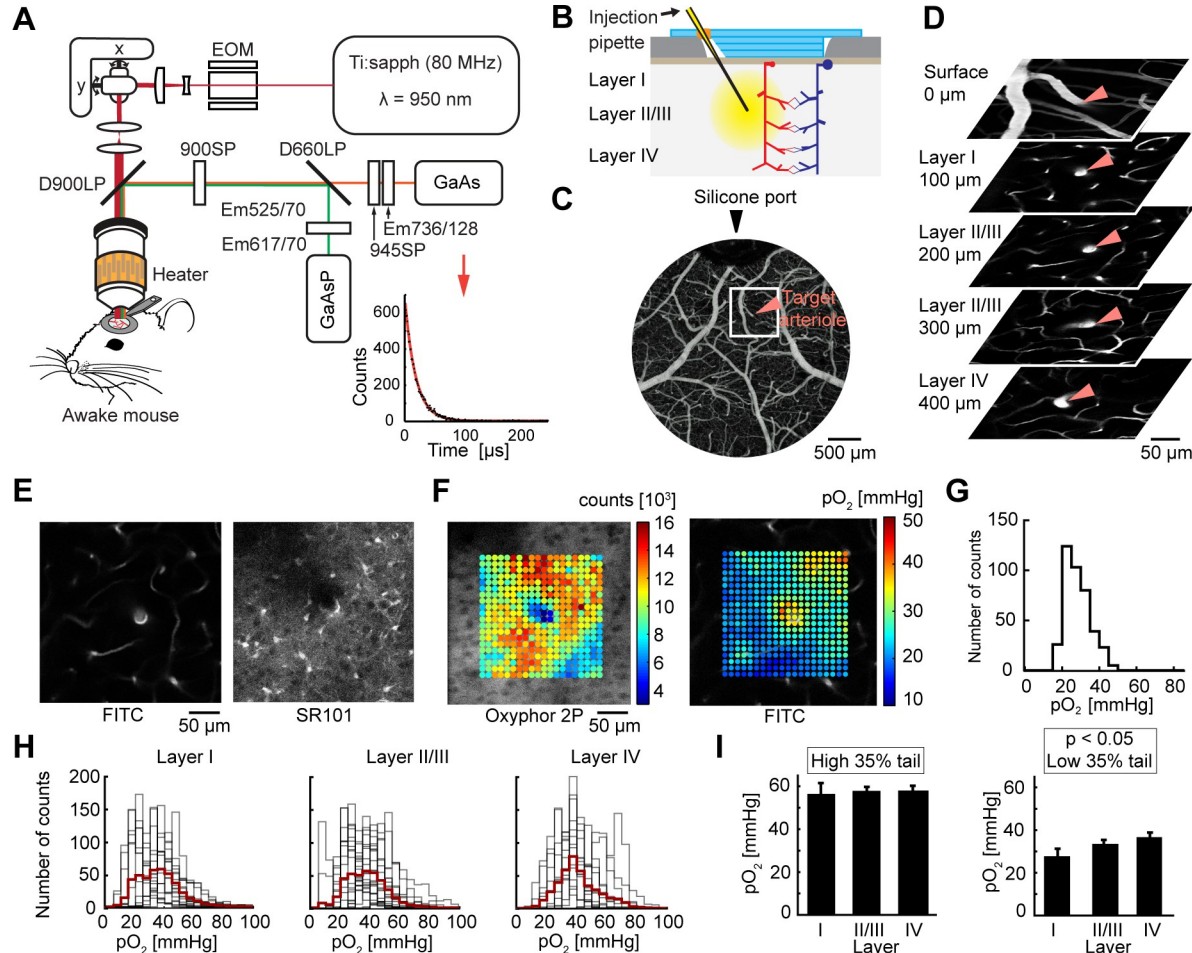

**Fig 1. Measurements of tissue partial pressure of O2 (pO2) across cortical layers of awake mice using 2-photon phosphorescence lifetime microscopy (2PLM).** (A) Imaging setup for 2PLM in awake, head-restrained mice. Ti:sapph (80 MHz)—femtosecond pulsed laser tuned to 950 nm; EOM—electro-optic modulator; D900LP and D660LP—long-pass dichroic mirrors with a cutoff at 900 and 660 nm, respectively; GaAsP—photomultiplier tubes; 900SP and 945SP—short-pass optical filters with a cutoff at 900 and 945 nm, respectively; Em525/70, Em617/70, Em736/128—bandpass emission filters. The inset in the lower right corner illustrates a phosphorescence decay; data (black) and fit (red) are overlaid. (B) Schematics of the chronic cranial window with a silicone port for intracortical injection of Oxyphor 2P. (C) An image of surface vasculature calculated as a maximum intensity projection (MIP) of a 2-photon image stack 0–300 µm in depth using a 5× objective. Individual images were acquired every 10 µm. Fluorescence is due to intravascular fluorescein isothiocyanate (FITC). Scale bar = 500 µm. (D) An example set of images tracking a diving arteriole throughout the top 400 µm of cortex (red arrowheads). Fluorescence is due to intravascular FITC. Scale bar = 50 µm. (E) A measurement plane 200 µm deep including intravascular FITC (left) and sulforhodamine 101 (SR101)–labeled astrocytes (right) for the same arteriole as in (D). Scale bar = 50 µm. (F) A square measurement grid of 20 × 20 points obtained from the imaging plane shown in (E). Left: photon counts are overlaid on the image of phosphorescence. Right: calculated pO2 values superimposed on a vascular FITC image. Scale bar = 50 µm. (G) Histogram of pO2 values corresponding to (F). (H) pO2 histograms across cortical layers for all 11 arterioles from 8 animals. Each panel shows overlaid histograms from each measurement plane (gray) and superimposed average (red). (I) Quantification of the top and bottom 35% of the pO2 distributions from (H). Error bars show standard error calculated using a mixed effects model implemented in R ($p > 0.1$ for the top 35% and $p < 0.05$ for the bottom 35%). Numerical values for (G–I) are provided in S1 Data (sheets 1G–1I).

radial grids of 400 points in the *XY* plane covering an area around a diving arteriole up to approximately 200 µm in radial distance. Overall, pO2 measurements were acquired at 51 planes along 11 diving arterioles in 8 subjects.

We traced individual diving arterioles from the cortical surface and acquired grids of pO2 points with an arteriole at the center (Figs 1D and S2). Fig 1E shows an example imaging plane crossing a diving arteriole. For each plane, a corresponding "reference" vascular image of intravascular FITC fluorescence was acquired immediately before and immediately after the

$pO_2$ measurements for co-registration of the measurement points in the coordinate system of the vascular network. The measurements are color-coded according to the $pO_2$ level (in mm Hg) and superimposed on the reference vascular image (Fig 1F).

We grouped $pO_2$ measurements according to the following depth categories: layer I (50–100 μm), layer II/III (150–300 μm), and layer IV (320–500 μm). Fig 1G shows the distribution of tissue $pO_2$ values for the specific example shown in Fig 1E and 1F, and Fig 1H shows similar histograms for the overall dataset pooled across subjects. The high tail of the overall $pO_2$ distribution (the upper 35%), reflecting oxygenation in tissue in the immediate vicinity of arterioles, did not change significantly as a function of depth (Fig 1I). This lack of dependence of the periarteriolar $pO_2$ on depth is in agreement with recent depth-resolved intravascular $pO_2$ studies in awake mice by us and others showing only a small decrease in the intravascular $pO_2$ along diving arteriolar trunks [17–19]. In contrast, there was a significant increase in the low tail (the bottom 35%) of the $pO_2$ distribution with depth (Fig 1I, $p < 0.05$), presumably indicating an increase in oxygenation of tissue in between capillaries in layer IV compared to layer I (see also S3 Fig). This depth-dependent increase questions the common believe that layer IV has higher baseline $CMRO_2$ compared to other layers. In addition, the median of the $pO_2$ distribution shifted to the higher $pO_2$ values below 200 μm (Fig 1H). The median increase can be explained, at least in part, by branching: At certain depths, diving arterioles branched, leading to the presence of highly oxygenated vessels (arterioles and capillaries) in the vicinity of diving arteriolar trunks.

## ODACITI model for CMRO2 extraction

To quantify the laminar profile of $CMRO_2$ in mouse cerebral cortex, we revised our model for estimation of $CMRO_2$ based on periarteriolar $pO_2$ measurements. The vascular architecture in the cerebral cortex features the absence of capillaries around diving arterioles [16,20,21]. Previously, we and others have argued that this organization agrees with the Krogh–Erlang model of $O_2$ diffusion from a cylinder [13], where a diving arteriole can be modeled as a single $O_2$ source for the tissue in the immediate vicinity of that arteriole where no capillaries are present. Indeed, radial $pO_2$ gradients around cortical diving arterioles in the rat SI approached 0 at the tissue radii corresponding to the first appearance of capillaries, and application of this model to estimate $CMRO_2$ yielded physiologically plausible results [20]. In the mouse, however, capillary-free periarteriolar spaces are smaller compared to those in the rat, and $O_2$ delivery by the capillary bed cannot be ignored [22]. Therefore, we revised the model to allow both arteriolar and capillary delivery.

In the Krogh–Erlang model, a blood vessel—approximated as an infinitely long cylinder with the radius $r = R_{ves}$—has a feeding tissue territory with a radius $r = R_t$. Inside this cylinder, all $O_2$ is derived from the vessel at the center. Therefore, according to the Krogh–Erlang model, tissue $pO_2$ will monotonically decrease with $r$ while moving away from the vessel until we reach $R_t$, where the total $O_2$ flux through the cylindrical surface with radius $R_t$ is 0 ($dpO_2/dr = 0$). Outside of this cylinder, the predicted $pO_2$ for $r>R_t$ has no physiological meaning as the model is defined for $r{\le}R_t$. Previously, we treated $R_t$ as the radius of the periarteriolar space devoid of capillaries [20].

The new model, which we call ODACITI ($O_2$ Diffusion from Arterioles and Capillaries Into Tissue), does not impose a restriction that tissue inside the $r < R_t$ radius is strictly supplied by $O_2$ from the center arteriole. Rather, it allows a "transitional region" inside the $R_t$ cylinder (at $R_0 < r < R_t$) that is strictly supplied by the capillary bed. Because $R_t$ is the radius of capillary-free periarteriolar space, the transitional region is supplied by the capillary bed but contains no capillaries. Consequently, there is a cylindrical boundary at $r = R_0$ inside $R_t$ through which the net $O_2$ flux is 0 (Fig 2A). For any cylindrical surface with radius $r$ between

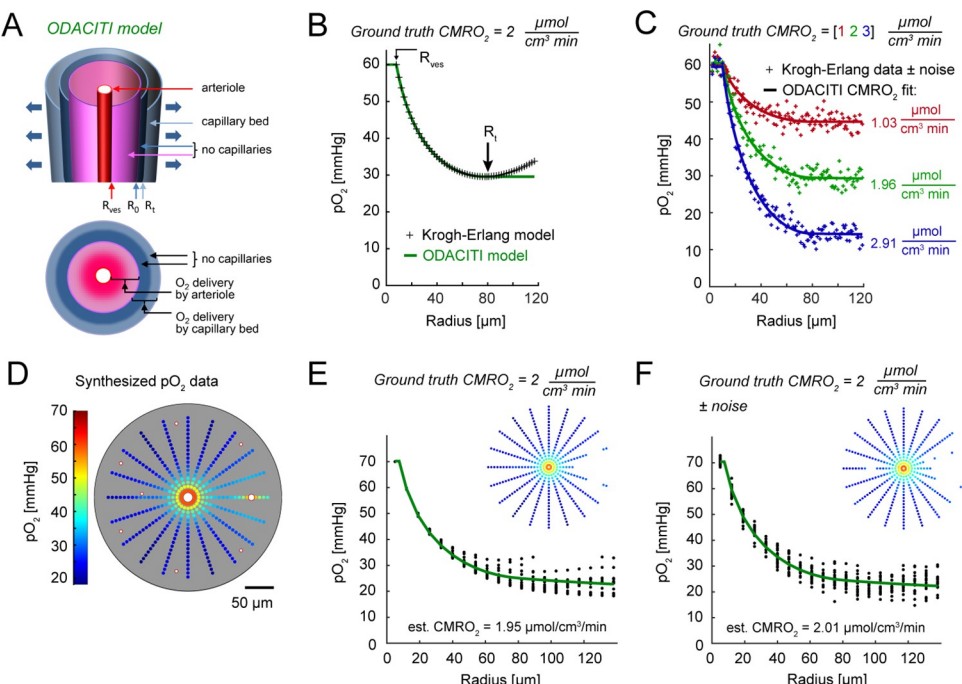

**Fig 2. ODACITI model and validation with synthetic data.** (A) Schematic illustration of the model assumptions. The center arteriole is the only source of $O_2$ for the pink periarteriolar region void of capillaries extending out to radius $r = R_t$. For $r > R_t$, delivery and consumption are balanced. (B) Comparison of the functional form of the Krogh–Erlang (black) and ODACITI model (green). The Krogh–Erlang model but not ODACITI forces an increase in partial pressure of $O_2$ ($pO_2$) beyond $R_t$. (C) Application of ODACITI to synthetic data with added Gaussian noise ($\sigma = 2$). In these data, $pO_2 = P_{ves}$ for $r < R_{ves}$ and $pO_2 = pO_2(R_t)$ for $r > R_t$. For $R_{ves} < r < R_t$, we solved for $pO_2$ using the Krogh–Erlang equation. Three cases with cerebral metabolic rate of $O_2$ (CMRO$_2$) of 1, 2, and 3 $\mu$mol cm$^{-3}$ min$^{-1}$ (color-coded) are superimposed. For each case, the ODACITI fit (solid line) is overlaid on the data points. (D) Simulated $pO_2$ data generated by solving the Poisson equation in 2D for a given geometry of vascular $O_2$ sources, including 1 highly oxygenated vessel (on the right) and given CMRO$_2$ = 2 $\mu$mol cm$^{-3}$ min$^{-1}$. (E) The $pO_2$ gradient as a function of the distance from the center arteriole. The green line shows ODACITI fit. (F) As in (E) after adding Gaussian noise ($\sigma = 2$ SD). Numerical values for (B, C, E, and F) are provided in S1 Data (sheets 2B, 2C, 2E, and 2F).

$R_{ves}$ and $R_t$, the total $O_2$ flux in ODACITI represents a difference between $O_2$ diffused from the vessel out and $O_2$ consumed in the inner tissue cylinder (see Methods). ODACITI solution is also defined for $r > R_t$, where CMRO$_2$ is exactly equal to the $O_2$ delivery rate of the capillary bed. In this way, ODACITI allows including data points beyond $R_t$, improving the robustness of the fitting procedure, and can better account for the influence of the capillary bed as compared to the Krogh–Erlang model (see Methods).

## Validation of the ODACITI model using synthetic data

To validate the ODACITI model, we used synthetic data where the ground truth (space-invariant) CMRO$_2$ was preset and thus known. First, we compared the functional form of the radial $pO_2$ gradient obtained with ODACITI to that obtained with the Krogh–Erlang equation (Fig 2B). To that end, we generated synthetic data by analytically solving the Krogh–Erlang and ODACITI equations for a given CMRO$_2$ and $R_t$. As expected, the models agree for $r < R_t$, producing the same monotonic decrease in $pO_2$ with increasing distance from the center arteriole. Beyond $R_t$, the Krogh–Erlang formula produces an unrealistic increase in $pO_2$, while ODACITI is able to take into account measured tissue $pO_2$ in the capillary bed and produces a more realistic, nearly constant level of $pO_2$ (Fig 2B). Next, we tested ODACITI on synthetic

data with added noise to mimic the noise of experimental measurements. To that end, we (i) amended the Krogh–Erlang solution by imposing a constant $pO_2$ equal to $pO_2(R_t)$ for $r > R_t$ to represent the capillary bed and (ii) added Gaussian noise ($\sigma = 2$). With these synthetic data, we were able to sufficiently recover the true $CMRO_2$ with ODACITI (Fig 2C). In another test, we performed fitting while selecting a constant $R_t$ between 60 and 80 μm. This resulted in an 18% error in the estimated $CMRO_2$ (S4 Fig). In comparison, the same variation in $R_t$ resulted in a 45% error while using a fitting procedure developed by Sakadzic et al. [20] to fit for $CMRO_2$ based on the Krogh–Erlang model. Thus, ODACITI is less sensitive to the error in $R_t$ compared to the Krogh–Erlang model.

ODACITI assumes that the first spatial derivative $dpO_2(r)/dr$ monotonically decreases in the proximity of the arteriole and remains near 0 in the capillary bed. In reality, however, this derivative varies in different directions from the center arteriole due to asymmetry of the vascular organization. Moreover, for some radial directions, $pO_2$ may remain high due to the presence of another highly oxygenated vessel (a small arteriolar branch or highly oxygenated capillary) [17,20]. To model this situation, we placed additional $O_2$ sources representing highly oxygenated blood vessels around the center arteriole (Fig 2D). When multiple nearby vessels serve as $O_2$ sources, no analytical solution for the tissue $pO_2$ map is available. Therefore, we solved the Poisson equation, which relates $CMRO_2$ and tissue $pO_2$, by means of finite element numerical modeling implemented in the software package FEniCS [23] (see Methods). Previously, we verified this implementation (for simple vascular geometries) by comparing the result to that of the Krogh–Erlang equation [24]. Fig 2D illustrates an example FEniCS output for a center 15-μm-diameter "arteriole" surrounded by seven 5-μm-diameter "capillaries" located 80–130 μm away from the diving arteriole (Fig 2D), reflecting the typical size of the region around diving arterioles void of capillaries in mouse cerebral cortex. The intravascular $pO_2$ was set to 70 mmHg and 35 mmHg for the arteriole and capillaries, respectively. In addition, we placed another "arteriole" with intravascular $pO_2$ set to 60 mmHg 110 μm away from the center arteriole to simulate the occasional presence of highly oxygenated vessels within our measurement grid.

These simulated data were used to devise a procedure for segmenting a region of interest (ROI) consistent with the assumption that $pO_2$ should decrease monotonically from the center arteriole until reaching a certain steady-state value within the capillary bed (see Methods). We then used the data included in the ROI to extract the $pO_2$ profile as a function of the radial distance from the center arteriole. Finally, we applied the ODACITI model to these radial $pO_2$ profiles to calculate $CMRO_2$. We calculated $CMRO_2$ in a noise-free case as well as with additive Gaussian noise (see Methods). In both cases, the model was able to recover the true $CMRO_2$ (Fig 2E and 2F).

## Estimation of $CMRO_2$ across cortical layers

Following validation with synthetic data (Fig 2), we used ODACITI to calculate $CMRO_2$ across the cortical depth, keeping the same depth categories as in Fig 1. For each imaging plane, we registered a $pO_2$ measurement grid with the corresponding vascular reference image. As with synthetic data, we segmented an ROI for each measurement grid and collapsed the data within each ROI into a corresponding radial $pO_2$ gradient (Fig 3A; see Methods). These periarteriolar $pO_2$ gradients are consistent with our previous measurements in the rat cortex under α-chloralose anesthesia [25] and those in a more recent study in the superficial cortex of awake mice [26]. Next, we performed fitting of the data to the ODACITI solution to quantify $CMRO_2$ (Fig 3B) (see Methods). Fig 3C shows depth-resolved $CMRO_2$ profiles along 11 diving arterioles in 8 subjects. We used a mixed effects model implemented in R to quantify $CMRO_2$ across the

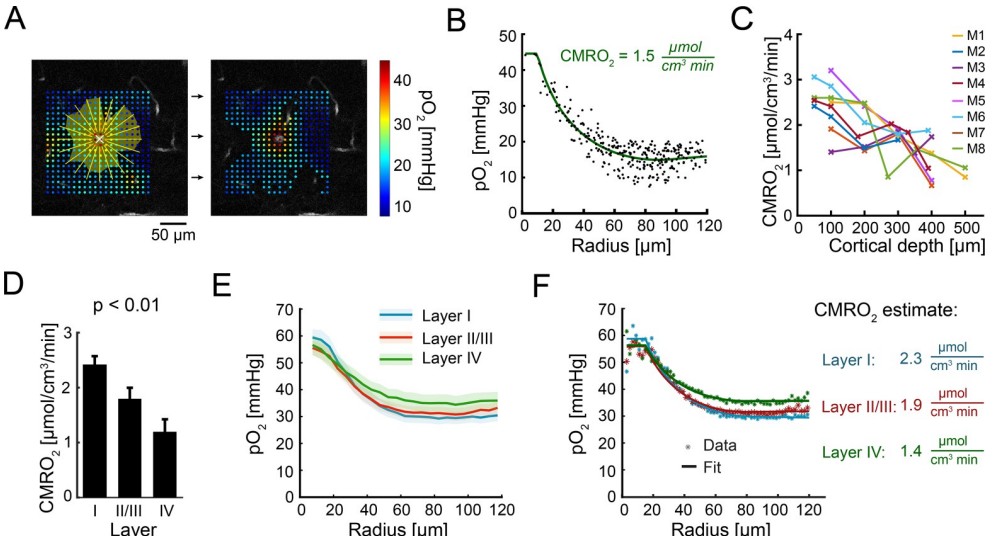

**Fig 3. Cerebral metabolic rate of $O_2$ (CMRO$_2$) estimation across cortical layers.** (A) An imaging plane 100 μm below the surface; a grid of partial pressure of $O_2$ (pO$_2$) points is overlaid on the vascular image. pO$_2$ values were interpolated along the radial yellow lines. The yellow shaded area extends along each direction until the first derivative becomes 0. The segmented region of interest (ROI) used for extraction of the radial pO$_2$ profile is shown on the right. This ROI also includes values within the low tail of the pO$_2$ distribution. (B) The radial pO$_2$ profile extracted from (A). The ODACITI fit is overlaid on the data. The fitted CMRO$_2$ value is 1.5 μmol cm$^{-3}$ min$^{-1}$. (C) Estimated CMRO$_2$ for the entire dataset of 51 planes along 11 diving arterioles in 8 subjects. Measurements along the same arteriole are connected with a line. Subjects are color-coded. (D) Quantification of CMRO$_2$ across layers using the data in (C). Error bars show standard error calculated using a linear mixed effects model implemented in R. (E) Experimentally measured tissue pO$_2$ (mean ± SEM, binned at 5 μm) across cortical layers (color-coded) as a function of distance from the arteriole. Each profile represents a grand average across subjects. (F) ODACITI fit for the data shown in (E) (binned at 2 μm). Numerical values for (B–F) are provided in S1 Data (sheets 3B–3F).

layers. This analysis revealed a significant reduction of CMRO$_2$ with depth ($p < 0.01$; Fig 3D). For this calculation, $R_t$ was fixed at 80 μm (see Methods).

For each arteriole, we also acquired pO$_2$ measurements at the cortical surface (S5 Fig). The surface measurements, however, were not used for estimation of CMRO$_2$ because of violation of the axial symmetry assumption (i.e., absence of cerebral tissue above the imaging plane).

Graphically, CMRO$_2$ scales with the steepness of the descent of the periarteriolar radial pO$_2$ profile [13,16,20,24]. In Fig 3E, we overlaid grand averaged radial profiles for each layer. Each curve was obtained by averaging all data for that layer (S6 Fig). As can be appreciated by visual inspection, the gradient corresponding to layer I has the steepest descent, while the descent of gradient in layer IV is more relaxed. This indicates that CMRO$_2$ in layer IV is lower compared to the upper layers, which is consistent with the model-based estimation shown in Fig 3D.

Finally, we performed post-mortem fluorescent labeling for COX in 4 additional subjects (see Methods). In agreement with the literature, our results show an increase in COX density from layer I to layer IV (S7 Fig).

Taken together, our results indicate that an increase in the low tail of the pO$_2$ distribution with depth—which likely reflects an increase in oxygenation of tissue between capillaries in layer IV compared to upper layers (Fig 1H and 1I)—can be explained at least in part by a depth-dependent decrease in CMRO$_2$ from layer I to layer IV (Fig 3D–3F). This is in contrast to the well-established fact that capillary, COX, and mitochondrial densities in the mouse SI all peak in layer IV [27,28] (see also S7 Fig), implying that none of these densities can serve as a proxy for oxidative energy metabolism under baseline (unstimulated) conditions.

## Discussion

In this work, we have achieved depth-resolved tissue $pO_2$ measurements in fully awake mice using 2PLM in combination with the $O_2$-sensitive probe Oxyphor 2P. We devised the ODACITI model for estimation of $CMRO_2$ from periarteriolar $pO_2$ gradients accounting for both arteriolar and capillary $O_2$ supply. We validated the model using synthetic data and then applied it to estimate the laminar $CMRO_2$ profile during the baseline level of neuronal activity (i.e., in the absence of external stimulation). Our results demonstrate a significant reduction of $CMRO_2$ with depth from layer I to layer IV. While these results are based on a model and its specific assumptions, they strongly suggest that, at the very least, the baseline $CMRO_2$ in layer IV does not exceed that in the upper cortical layers.

In cerebral cortex, neuronal cell type distribution as well as cellular, synaptic, and microvascular densities vary between cortical layers [1,4,17,28–33]. Cortical laminar architecture is also readily revealed by counting mitochondria [27] or staining for COX—the last enzyme in the respiratory electron transport chain located in the inner mitochondrial membrane [3,4]. In SI, COX distribution features a high-density band in cortical layer IV, which has been hypothesized to reflect higher $CMRO_2$ compared to other layers. Further, COX was found to have a strong correlation with microvascular but not synaptic density—a finding that was interpreted to reflect high layer IV $CMRO_2$ in the "idling state" [4], where about half of the energy expenditure reflects processes other than synaptic and spiking activity [34,35]. The present measurements and calculations do not support the idea of high baseline $CMRO_2$ in layer IV.

If this is the case, what is the purpose of high COX density in layer IV? In our recent study, we found that intravascular $pO_2$ changes ($\Delta pO_2$) in response to a sensory stimulus across layers were conserved [18], indicating a conserved ratio between demand and supply. Taken together with existing data on larger stimulus-induced increases in cerebral blood flow (CBF) ($\Delta CBF$) in layer IV compared to other layers [36], this finding implies that the laminar $\Delta CMRO_2$ profile should also peak in layer IV in order for $\Delta pO_2$ to remain invariant. Thus, COX may reflect layer-specific differences in peak energetic demands during transient neuronal signaling events. These events may include not only task-induced computation performed by local neuronal circuits (e.g., the barrel cortex response to a whisker touch [31]) but also other dynamic neuronal processes occurring on larger spatiotemporal scales such as neuromodulation and sleep [37].

In this study, we observed a shift towards higher oxygenation of tissue within the capillary bed—quantified as the mean of the low tail of $pO_2$ distribution—with increasing depth. Because our $pO_2$ grids were always placed around diving arterioles, the low tail of the $pO_2$ distribution may not reflect oxygenation of the capillary bed remote from arteriolar $O_2$ sources. This observation, however, is potentially in line with our recent intravascular $pO_2$ study in awake mice, where we showed that mean capillary $pO_2$ in layer IV was approximately 15% higher than that in the upper layers [17]. It is also consistent with a recent study where tissue $pO_2$ measurements were performed at different depths while a Clark-type polarographic electrode was advanced through the cortical layers [38]. We speculate that higher baseline oxygenation in layer IV may serve as an $O_2$ reserve for transient increases in neuronal activity. Beyond this reserve, upregulation in aerobic glycolysis may play a role alongside oxidative phosphorylation to rapidly supply ATP [39–41].

In principle, higher capillary density in layer IV [28] may effectively shrink the periarteriolar cylinder where all $O_2$ is provided by the diving arteriole. This hypothetical scenario, however, would not affect the second derivative of the periarteriolar $pO_2$ gradient (which reflects $CMRO_2$). Rather, it would decrease the radius of this cylinder, $R_t$. Although in our calculations of $CMRO_2$ $R_t$ was a fixed parameter, we show that the sensitivity of ODACITI to varying $R_t$

was relatively low. In addition, allowing $R_t$ to vary while fitting for CMRO$_2$ did not reveal depth dependence.

The current study is part of our ongoing effort to improve 2PLM [10,14,17,18,20,25,42] and microscopic estimation of CMRO$_2$ [20,24,43]. In principle, if we knew space-resolved tissue pO$_2$ as well as intravascular pO$_2$ for each blood vessel within that tissue, we could solve for CMRO$_2$ (which may vary in space and time) by finding an inverse solution for the Poisson diffusion equation [24]. In practice, however, the signal-to-noise ratio (SNR) of our measurements may not be sufficient to accurately resolve pO$_2$ gradients around capillaries. In addition, our current spatial resolution limits simultaneous measurements of capillary and tissue pO$_2$, and sampling pO$_2$ in 3D also remains a challenge. To mitigate these limitations, similar to the Krogh–Erlang model, the ODACITI model relies on radial pO$_2$ gradients around diving arterioles, assuming cylindrical symmetry. In reality, highly oxygenated arteriolar branches and/or low-branching-order capillaries can often be present on one side of a diving arteriole, requiring ROI segmentation. Applied to synthetic data, our data analysis stream—including the segmentation algorithm followed by ODACITI estimation—was able to recover the "ground truth" CMRO$_2$, validating the method. In the future, further improvements in pO$_2$ probes and 2PLM technology may enable estimation of CMRO$_2$ in 3D using numeric methods without the need to segment the data [24].

The CMRO$_2$ estimate derived from a periarteriolar region is likely to represent CMRO$_2$ in the surrounding tissue fed by capillaries as well as in neighboring perivenular regions, as long as we stay within the same cortical column and layer. Several convergent lines of evidence support this hypothesis. First, it has been shown that the location of penetrating arterioles does not map onto the structure of cortical columns and septa in the barrel cortex [28,44]. Second, the mitochondrial/cytochrome density within a column does not co-vary with distance from penetrating arterioles [45]. Third, quantification of mitochondrial ATP production as a function of pO$_2$ [46] suggests that the production should not be affected within the range of tissue pO$_2$ values reported here as well as in previous studies in healthy adult mice [19,26]. Fourth, a microscopic reconstruction study has concluded that neuronal somas and vascular densities do not track each other on the 100-µm scale of cortical lamina [47]. Further, O$_2$ consumption occurs locally at mitochondria that are present not only in somas but also along the dendrites that often extend through the width of a cortical column and beyond [29,48]. Therefore, even when a soma happens to be located next to an arteriole, the dendrites can span a large volume encompassing the capillary bed and perivenular regions. Finally, formation of neuronal circuits during development is not constrained by the location of feeding arterioles [49], and the baseline O$_2$ extraction fraction in the SI of awake adult mice is only 30%–40%, leaving a safe margin for additional extraction from capillaries and venules during neuronal activation even between downstream capillaries and in the perivenular region [17]. Collectively, this body of evidence provides a strong scientific premise for our assumption of space-invariant CMRO$_2$ at the local scale of cortical columns and lamina.

Our measurements were limited to cortical layers I–IV. In 2-photon microscopy, depth penetration is fundamentally limited by out-of-focus excitation [50]. Because light is scattered and absorbed by tissue, the laser power must increase with depth to overcome these effects and deliver a sufficient number of photons to the focal volume. In our experiments, an increase in the number of scattered photons would increase the probability of out-of-focus excitation of Oxyphor 2P, generating background phosphorescence. The lifetime of this background phosphorescence would report pO$_2$ at locations outside the focal volume, contributing to experimental error and smoothing out periarteriolar pO$_2$ gradients. In this study, we acquired data using 2 different objectives with numerical aperture (NA) = 0.5 and NA = 1. While higher NA reduced out-of-focus excitation (quantified as signal-to-background ratio ([SBR] in S8 Fig),

we observed the same behavior of $CMRO_2$ irrespective of the NA. Therefore, we believe that the smoothing effect was not significant. In the future, a combination of 2PLM and light sculpting techniques [51] could be used to experimentally measure the amount of generated out-of-focus signal. 2PLM can also be implemented using two laser beams of different color that can be displaced in space, minimizing out-of-focus excitation [52,53].

This study was performed using fully awake mice with no anesthesia or sedation. While our mice were well trained to accept the head restraint and habituated to the imaging environment, we cannot rule out upregulation of adrenergic activity associated with arousal and/or stress. Previous studies have shown that aerobic glycolysis and glycogenolysis are triggered by activation of β1 and β2 adrenergic receptors, respectively [39,54,55]. Both of these processes are layer-specific in cerebral cortex. Therefore, the relative contribution of oxidative and glycolytic pathways to overall energy metabolism across cortical layers can depend on the state of adrenergic neuromodulation [56]. In the future, a combination of 2PLM and imaging of novel genetically encoded probes for norepinephrine and other neuromodulators [57,58] should allow quantification of $CMRO_2$ in the context of different brain states.

Knowing layer-specific $CMRO_2$ in cerebral cortex is important for better understanding of normal brain physiology as well as pathophysiology in diseases that affect cerebral microcirculation [59–63]. Baseline $CMRO_2$ also affects blood-oxygenation-level-dependent (BOLD) functional magnetic resonance imaging (fMRI) signals, altering the hemodynamic response for the same neuronal reality [43,64–67]. In the future, extending our approach to stimulation studies would help the informed interpretation and modeling of high-resolution BOLD signals enabled by recent technological developments [68–71], bringing noninvasive imaging one step closer to the spatial scale of local neuronal circuits [70,72–78].

## Methods

### O$_2$ probe

The synthesis and calibration of the $O_2$ probe Oxyphor 2P were performed as previously described [14]. The phosphorescence lifetime of Oxyphor 2P is inversely proportional to the $O_2$ concentration. Compared to its predecessor, PtP-C343 [12], the optical spectrum of Oxyphor 2P is shifted to the longer wavelengths, with the 2-photon absorption and phosphorescence peaks at 960 nm and 760 nm, respectively, enabling deeper imaging. Other advantages include a large 2-photon absorption cross section (approximately 600 GM near 960 nm) and higher phosphorescence quantum yield (0.23 in the absence of $O_2$). In addition, the phosphorescence decay of Oxyphor 2P can be more closely approximated by a single exponential function. Calibration for conversion of the phosphorescence lifetime to $pO_2$ was obtained in independent experiments, where $pO_2$ was detected by a Clark-type electrode as previously described [14].

### Two-photon imaging

Images were obtained using an Ultima 2-photon laser scanning microscopy system (Fig 1A) (Bruker Fluorescence Microscopy). Two-photon excitation was provided by a Chameleon Ultra femtosecond Ti:Sapphire laser (Coherent) tuned to 950 nm.

For 2-photon phosphorescence lifetime microscopy (2PLM), laser power and excitation gate duration were controlled by 2 electro-optic modulators (EOMs) (350-160BK, Conoptics) in series, with an effective combined extinction ratio > 50,000. This was done to mitigate bleed-through of the laser light and reduce the baseline photon count (i.e., during gate off periods), which was critical for accurately fitting phosphorescence decays.

We used a combination of a Zeiss 5× objective (Plan-Neofluar, NA = 0.16) for coarse imaging and an Olympus 20× objective (UMPlanFI, NA = 0.5) for fine navigation under the glass window. In 4 subjects, the same 20× objective was used for 2PLM. In an additional 4 subjects, a modification of the headpost allowed using a wider 20× objective with higher NA (XLUM-PlanFLNXW, NA = 1.0). Data obtained with NA = 1.0 had higher signal-to-background ratio (SBR) (S8 Fig). An objective heater (TC-HLS-05, Bioscience Tools) was used to maintain the temperature of the water between the objective lens and cranial window at 36.6˚C to avoid the objective acting as a heat sink and to comply with the Oxyphor 2P calibration temperature [17,79,80].

The emission of Oxyphor 2P was reflected with a low-pass dichroic mirror (900-nm cutoff wavelength; custom-made by Chroma Technologies), subsequently filtered with a 795/150-nm emission filter (Semrock), and detected using a photon counting photomultiplier tube (PMT; H7422P-50, Hamamatsu). A second PMT operated in analog mode (H7422-40, Hamamatsu) with a 525/50-nm or 617/73-nm emission filter (Semrock) was used to detect fluorescein iso-thiocyanate (FITC)-labeled dextran or the astrocytic marker sulforhodamine 101 (SR101), respectively (see "Delivery of optical probes" below). A 945-nm short-pass filter (Semrock) was positioned in front of the PMTs to further reject laser illumination.

At each imaging plane up to 400 μm below the cortical surface, an approximately $300 \times 300$ μm field of view (FOV) was selected for $pO_2$ measurements. At each plane, $pO_2$ measurements were performed serially arranged in a square or radial grid of 400 points.

The phosphorescence was excited using a 13-μs-long excitation gate, and the emission decay was acquired during 287 μs. The photon counts were binned into 2-μs-long bins. Typically, 50 decays were accumulated at each point, with a total acquisition time of 15 ms per point. With 400 points per grid, 1 grid was acquired within 6 s. All points were revisited 20 times, yielding a total of 1,000 excitation cycles per point (50 cycles × 20 repetitions) acquired within 120 s. Dividing the acquisition of 1,000 cycles per point in blocks of 20 repetitions increased tolerance against motion (see "Motion correction" below) at a price of a slightly reduced sampling rate due to the settling time of the galvanometer mirrors while moving from point to point.

## Implantation of the cranial window and headpost

All animal procedures were performed in accordance with the guidelines established by the Institutional Animal Care and Use Committee (IACUC) at Boston University (protocol PROTO202000040) and University of California San Diego (protocol S14275). Both protocols adhered to the US Government Principles for the Utilization and Care of Vertebrate Animals Used in Research, Teaching, and Testing.

We used 22 adult C57BL/6J mice of either sex for $pO_2$ measurements (age: 4–7 months). Fourteen of them were rejected due to imperfect healing of the cranial window or unsuccessful intracranial injections, and 8 were used for $pO_2$ measurements. An additional 4 adult C57BL/6J mice were used for immunocytochemistry. Mice had free access to food and water and were held in a 12 h light/12 h dark cycle.

The surgical procedure for implantation of a chronic optical window was performed as previously described [81,82]. Briefly, dexamethasone was injected approximately 2 h prior to surgery. Mice were anesthetized with ketamine/xylazine or isoflurane (2% in $O_2$ initially, 1% in $O_2$ for maintenance) during surgical procedures; their body temperature was maintained at 37˚C. A 3-mm cranial window with a silicone port was implanted over the left barrel cortex, and the headpost was mounted over the other (right) hemisphere. The glass implant contained a hole, filled with silicone [82,83], allowing intracortical injection of the $O_2$ probe (Figs 1B and

S1). The window and the headpost were fixed to the skull in a predetermined orientation such that, when the mouse head was immobilized in the mouse holder ("hammock"), the window plane would be horizontal. Dextrose saline (5% dextrose, 0.05 ml) was injected subcutaneously before discontinuing anesthesia. Post-operative analgesia was provided with buprenorphine (0.05 mg/kg subcutaneously) injected approximately 20 min before discontinuing anesthesia. A combination of sulfamethoxazole/trimethoprim (Sulfatrim) (0.53mg/mL sulfamethoxazole and 0.11mg/mL trimethoprim) and ibuprofen (0.05 mg/ml) was provided in drinking water starting on the day of surgery and for 5 d after surgery. Generally, full recovery and return to normal behavior were observed within 48 h post-op.

### Delivery of optical probes

Oxyphor 2P and astrocytic marker SR101 were delivered by intracortical microinjection through the silicone port in the glass window (Figs 1B and S1). Oxyphor 2P was diluted to 340 μM in artificial cerebrospinal fluid (ACSF; containing 142 mM NaCl, 5 mM KCl, 10 mM glucose, 10 mM HEPES ([Na Salt], 3.1 mM $CaCl_2$, 1.3 mM $MgCl_2$ [pH 7.4]) and filtered through a 0.2-μm filter (Acrodisc 4602, PALL). SR101 (Sigma S359) was added for a final concentration of 0.2 mM. We used quartz-glass capillaries with filament (O.D. 1.0 mm, I.D. 0.6 mm; QF100-60-10, Sutter Instrument). Pipettes were pulled using a P-2000 Sutter Instrument puller. For each pulled pipette, the tip was manually broken under the microscope to obtain an outer diameter of 15–25 μm. A pipette was filled with a mixture of Oxyphor 2P and SR-101 (thereafter referred to as the "dye") and fixed in a micromanipulator at an angle of approximately 35˚. The pipette was guided under visual control through the silicone port to its final location in tissue while viewing the SR101 fluorescence through the microscope eyepiece. First, to target the tissue surrounding a diving arteriole, the pipette was oriented above the window such that its projection onto the window coincided with the line connecting the diving point with the silicone port. The pipette was then retracted and lowered to touch the surface of the port. There, a drop of the dye solution was ejected to ensure that the pipette was not clogged. A holding positive pressure was maintained (using PV830 Picopump, World Precision Instruments) to avoid clogging of the pipette while advancing through the port. An experimenter could recognize the pipette emerging below the port by the dye streaming from the tip. At this point, the holding pressure was quickly set to 0 in order to avoid spilling of the dye on the cortical surface. Next, the pipette was advanced for approximately 600 μm along its axis at 35˚. Below approximately 100 μm, some holding pressure was applied to allow leakage of the dye that was used to visualize the pipette advancement. The pressure was manually adjusted to ensure visible spread of the dye without displacing cortical tissue. When the target artery was reached, the pipette was withdrawn by approximately 50 μm, and holding pressure was maintained for approximately 20 min to allow slow diffusion of the dye into the tissue. A successful injection was recognized by the appearance of SR101 in the perivascular space around the targeted arteriole but not around its neighbors. The contrast between the targeted and neighboring arterioles was lacking when the dye was injected too shallow or too vigorously, in which case the experiment was aborted. After the loading, the pressure was set to 0, and the pipette was withdrawn. The mouse was placed back in its home cage to recover for approximately 40 min until the start of the imaging session. At that time, the dye had usually diffused within a 300 to 600 μm radius around the targeted arteriole.

Insertion of the pipette into the brain tissue could in principle induce cortical spreading depression (CSD). Although we think that this is an unlikely possibility because the pipettes were very thin, we did not measure a DC potential during the injection and cannot rule out that CSD occurred in some cases. Previous studies have shown that in mice, CSD results in a

transient CBF decrease that recovers within approximately 30 min [84,85]. Therefore, waiting for 40 min after the injection also ensured that CBF was back to its baseline even in the unlikely event of CSD.

FITC-dextran (FD2000S, 2 MDa, Sigma-Aldrich) was used to visualize cortical vasculature. Mice were briefly anesthetized with isoflurane, and 50 μl of the FITC solution (5% in normal saline) was injected retro-orbitally prior to intracortical delivery of Oxyphor 2P for each imaging session.

## Habituation to head fixation

Starting at least 7 d after the surgical procedure, mice were habituated in 1 session per day to accept increasingly longer periods of head restraint under the microscope objective (up to 1 h/ day). During the head restraint, the mouse was placed on a hammock. A drop of diluted sweetened condensed milk was offered every 20–30 min during the restraint as a reward. Mice were free to readjust their body position and from time to time displayed natural grooming behavior.

## Estimation of pO$_2$

All image processing was performed using custom-designed software in MATLAB (Math-Works). For each point, cumulative data from all excitation cycles available for that point were used for estimation of the phosphorescent decay. Starting 5 μs after the excitation gate to minimize the influence of the instrument response function, the decay was fitted to a single exponential function:

$$N(t) = N_0 e^{-t/\tau} + x, \tag{1}$$

where $N(t)$ is the number of photons at time $t$, $N_0$ is the number of photons at $t = 0$ (i.e., 5 μs after closing the excitation gate), $\tau$ is the phosphorescence lifetime, and $x$ is the offset due to non-zero photon count at baseline. The fitting routine was based on the nonlinear least squares method using the MATLAB function *lsqnonlin*. The phosphorescence lifetime $\tau$ was then converted into absolute pO$_2$ using an empirical biexponential form:

$$pO_2 = A_1 e^{-\tau/t1} + A_2 e^{-\tau/t2} + y_0 \tag{2}$$

where parameters $A_1$, $t_1$, $A_2$, $t_2$, and $y_0$ were obtained during independent calibrations [14].

The locations of pO$_2$ measurements were co-registered with FITC-labeled vasculature. Color-coded pO$_2$ values are overlaid on vascular images (grayscale) in all figures displaying pO$_2$ maps.

## Identification of data corrupted by motion

To exclude data with excessive motion, an accelerometer (ADXL335, Sainsmart, Analog Devices) was attached below the mouse hammock. The accelerometer readout was synchronized with 2-photon imaging and recorded using a dedicated data acquisition system (National Instruments). During periods with extensive body movement (e.g., grooming behavior), the accelerometer signal crossed a predefined threshold, above which data were rejected. In pilot experiments, in addition to the accelerometer, a webcam (Lifecam Studio, Microsoft; infrared filter removed) with infrared illumination (M940L3-IR [940 nm] LED, Thorlabs) was used for video recording of the mouse during imaging. The videos and accelerometer readings were in general agreement with each other. Therefore, accelerometer data alone were used to calculate the rejection threshold. Because every point was revisited 20 times, typically at least

10 repetitions (or 500 excitation cycles) were unaffected by motion and were used to estimate τ. The MATLAB function *lsqnonlin* used to fit phosphorescent decays returned the residual error for each fit, which we plotted against the number of cycles (S9 Fig). At around 500 cycles, the error stabilized at a low level. Therefore, we quantified points where at least 500 cycles were available.

## Estimation of CMRO$_2$

The O$_2$ transport to tissue is thought to be dominated by diffusion. In the general case, the relationship between pO$_2$, denoted as $P(\overrightarrow{r}, t)$, and the O$_2$ consumption rate, denoted as CMRO$_2(r,t)$, can be described by

$$\partial P(\overrightarrow{r}, t)/\partial t = D\nabla^2 P(\overrightarrow{r}, t) - \mathrm{CMRO}_2(\overrightarrow{r}, t)/\alpha, \tag{3}$$

where $\nabla^2$ is the Laplace operator in 3 spatial dimensions, and $D$ and $\alpha$ are the diffusion coefficient and solubility of O$_2$ in tissue, respectively. For all our calculations, we assumed $\alpha =$ 1.39 µM mmHg$^{-1}$ and D = $4 \times 10^{-5}$ cm$^2$ s$^{-1}$ [86]. This equation is only applicable outside the blood vessels supplying O$_2$ to the tissue. Oxygen supplied by a blood vessel is represented by a boundary condition of pO$_2$ imposed at the vessel wall. When the system is in steady state, the term $\partial P(\overrightarrow{r}, t)/\partial t$ can be neglected. If we also assume that there is no local variation of pO$_2$ in the vertical $z$-direction, that is, the direction along the cortical axis parallel to penetrating arterioles, Eq 3 simplifies to

$$\nabla^2 P(r) = \frac{\mathrm{CMRO}_2(r, t)}{D\alpha} \tag{4}$$

where $P(r)$ represents pO$_2$ measured at the radial location $r$ and $\nabla^2$ refers to the 2-dimensional Laplace operator.

Here, we derive a specific solution to the forward problem of this partial differential equation assuming a cylindrical central arteriole with radius $r = R_{\mathrm{ves}}$ and a region around the arteriole void of capillaries within a radius $r = R_{\mathrm{t}}$, similar to the Krogh–Erlang model of O$_2$ diffusion from a cylinder [13]. In contrast to the Krogh–Erlang model, we assume that, while most of the $R_{\mathrm{ves}} < r < R_{\mathrm{t}}$ region is supplied by the arteriole, there exists a transitional region inside the $R_{\mathrm{t}}$ cylinder (at $R_0 < r < R_{\mathrm{t}}$) that contains no capillaries but is supplied by the capillary bed. With these assumptions, we can derive an analytical solution to Eq 4 that we describe below.

We generally denote the partial differential equation in cylindrical coordinates as

$$\frac{1}{r}\frac{\partial}{\partial r}\left(r\frac{\partial \mathrm{pO}_2(r)}{\partial r}\right) = \bar{H}(r - R_{\mathrm{ves}}) - \bar{H}(r - R_{\mathrm{t}}), \tag{5}$$

where pO$_2(r)$ is the pO$_2$ as a function of the radial distance from the center of a diving arteriole, and $H(r)$ is a Heaviside step function $\bar{H}(r) = H(r)\mathrm{CMRO}_2/(D\alpha)$. Therefore, Eq 5 considers 3 spatial regions $r<R_{\mathrm{ves}}$, $R_{\mathrm{ves}} \leq r \leq R_{\mathrm{t}}$, and $r>R_{\mathrm{t}}$, where only in the middle region (e.g., for $R_{\mathrm{ves}} \leq r \leq R_{\mathrm{t}}$) the right-hand side of Eq 5 is non-zero. Additionally, we required that pO$_2(r)$ is a continuous function at $r = R_{\mathrm{t}}$ (e.g., $\lim_{r \to R_{\mathrm{t}}^-} \mathrm{pO}_2(r) = \lim_{r \to R_{\mathrm{t}}^+} \mathrm{pO}_2(r)$) and at $r = R_{\mathrm{ves}}$ (e.g., $\lim_{r \to R_{\mathrm{ves}}^+} \mathrm{pO}_2(r) = \mathrm{pO}_{2,\mathrm{ves}}$) and a smooth function at $r = R_{\mathrm{t}}$ (e.g., $\lim_{r \to R_{\mathrm{t}}^-} \partial\mathrm{pO}_2(r)/\partial r = \lim_{r \to R_{\mathrm{t}}^+} \partial\mathrm{pO}_2(r)/\partial r$).

Importantly, to better account for the O$_2$ delivery by the capillary bed that surrounds the arteriole, we assumed that $\partial\mathrm{pO}_2(r)/\partial r = 0$ for some $r = R_0$, where $R_{\mathrm{ves}}<R_0 \leq R_{\mathrm{t}}$. Finally, pO$_2$ values that we measured at $r<R_{\mathrm{ves}}$ did not show any diffusion gradients because they originated from extracellular dye in the tissue next to the vessel. Therefore, in our model we assumed that

$pO_2(r)_{r<R_{ves}} = pO_{2,ves}$. We considered the following solution of Eq 5, which satisfies the above conditions:

$$pO_2(r)_{r<R_{ves}} = pO_{2,ves} \tag{6}$$

$$pO_2(r)_{R_{ves} \leq r \leq R_t} = pO_{2,ves} + \frac{CMRO_2}{4D\alpha}\left(r^2 - R_{ves}^2 - 2R_{ves}^2 ln\frac{r}{R_{ves}}\right) + \beta ln\frac{r}{R_{ves}} \tag{7}$$

$$pO_2(r)_{r>R_t} = pO_{2,ves} + \frac{CMRO_2}{4D\alpha}\left(R_t^2 - R_{ves}^2 - 2R_{ves}^2 ln\frac{r}{R_{ves}} + 2R_t^2 ln\frac{r}{R_t}\right) + \beta ln\frac{r}{R_{ves}} \tag{8}$$

This model, which we nicknamed ODACITI (for "O$_2$ Diffusion from Arterioles and Capillaries Into Tissue"), was used to fit for CMRO$_2$, pO$_{2,ves}$, and β, given the arteriolar radius $R_{ves}$ and the radius of capillary-free periarteriolar space $R_t$. $R_{ves}$ was estimated from vascular FITC images as the full-width at half-maximum of the intensity profile drawn across the arteriole. $R_t$ was fixed to 80 μm based on observation of the capillary bed in mice (S10 Fig). To simplify the fitting procedure, parameter β was used in Eqs 6–8 to replace $CMRO_2(R_{ves}^2 - R_0^2)/(2D\alpha)$.

Notably, the ODACITI model behaves not necessarily the same at $R_t$ as the original Krogh–Erlang model given by

$$pO_2(r) = pO_{2,ves} + \frac{CMRO_2}{4D\alpha}\left(r^2 - R_{ves}^2 - 2R_t^2 ln\frac{r}{R_{ves}}\right) \tag{9}$$

Similar to the ODACITI model, this expression satisfies Eq 4 for $R_{ves} \leq r \leq R_t$. However, the Krogh–Erlang model requires that O$_2$ delivered by the vessel in the center is consumed by the tissue within the cylinder with the radius $R_t$, which implies that O$_2$ flux $\left(J(r) = -D\alpha\frac{\partial pO2(r)}{\partial r}\right)$ is equal to 0 at $r = R_t$. In the Krogh–Erlang model, the expression for the O$_2$ flux ($J_{KE}(r)$) for $R_{ves} \leq r \leq R_t$ is given by

$$J_{KE}(r) = CMRO_2(R_t^2 - r^2)/(2r) \tag{10}$$

and total O$_2$ flux through any cylindrical surface with length $\Delta z$ and radius $r$ from the vessel is equal to the product of the metabolic rate of O$_2$ and the volume of the tissue that remains outside the cylinder with radius $r$:

$$J_{KE}(r)2\pi r\Delta z = CMRO_2\pi(R_t^2 - r^2)\Delta z \tag{11}$$

If we denote the O$_2$ flux in ODACITI as $J_{OD}(r)$, total O$_2$ flux through the wall of the central arteriole, for the arteriolar segment of length $\Delta z$, is given by

$$J_{OD}(R_{ves})2\pi R_{ves}\Delta z = -D\alpha\beta 2\pi\Delta z = CMRO_2(R_0^2 - R_{ves}^2)\pi\Delta z \tag{12}$$

For any cylindrical surface with radius $r$ between $R_{ves}$ and $R_t$, the total O$_2$ flux in ODACITI is given by

$$J_{OD}(r)2\pi r\Delta z = -D\alpha\beta 2\pi\Delta z - CMRO_2\pi(r^2 - R_{ves}^2)\Delta z \tag{13}$$

which represents a difference between the oxygen diffused from the vessel out ($-D\alpha\beta 2\pi\Delta z$) and the oxygen consumed in the inner tissue cylinder ($CMRO_2\pi(r^2 - R_{ves}^2)\Delta z$). Finally, total O$_2$ flux through a cylindrical surface with $r>R_t$ is constant:

$$\begin{aligned}J_{OD}(r)2\pi r\Delta z J(r)2\pi r &= -D\alpha\beta_3 2\pi - CMRO_2\pi(r^2 - R_{ves}^2) \\ &= -D\alpha\beta 2\pi\Delta z - CMRO_2\pi(R_t^2 - R_{ves}^2)\Delta z\end{aligned} \tag{14}$$

which is in agreement with the model assumption that for $r > R_t$, $CMRO_2$ is exactly equal to the $O_2$ delivery rate of the capillary bed. In ODACITI, if oxygen flux at $r = R_t$ is 0 ($J_{OD}(R_t) = 0$), then $R_0 = R_t$, all oxygen delivered by the arteriole is consumed inside the $r < R_t$ tissue volume (e.g., $-D\alpha\beta 2\pi = CMRO_2\pi(R_t{}^2 - R_{ves}{}^2)$), and ODACITI scales back exactly to the Krogh–Erlang model with the addition that $pO_2$ is constant for $r < R_{ves}$ and $r > R_t$. However, in ODACITI, it is possible for the radius where $J_{OD}(r) = 0$ to be smaller than $R_t$, which can better account for the influence of the capillary bed. In addition, this model allows one to include more measured data points from small ($r < R_{ves}$) and large radii ($r > R_t$) into the fitting procedures as compared to using the Krogh–Erlang model.

Full derivation of the model is provided in S1 Appendix.

## Segmentation of regions of interest

ODACITI describes $pO_2$ gradients around diving arterioles for an ideal case of radial symmetry. In practice, however, capillary geometry around diving cortical arterioles is not perfectly radially symmetric. The model also assumes that $pO_2$ decreases monotonically with increasing radial distance from the arteriole until reaching a certain stable level within the capillary bed. In reality, however, at a certain distance $pO_2$ may increase again due to the presence of another highly oxygenated blood vessel that can be (a branch of) a diving arteriole or a post-arteriolar capillary. To mitigate this issue, we devised an algorithm for automated segmentation of a ROI for each acquired $pO_2$ grid. First, tissue $pO_2$ measurements were co-registered with the vascular anatomical images to localize the center of the arteriole. Next, 20 equally spaced radii were drawn in the territory around a center arteriole, and a $pO_2$ vector was interpolated as a function of the radial distance from the arteriole. These vectors were spatially filtered using the cubic smoothing spline *csaps* MATLAB function with smoothing parameter $p = 0.001$ to reduce the effect of noise in the measurements. For each vector, we calculated the first spatial derivative $\frac{dpO_2}{dr}$ and included all $pO_2$ data points with radii where $\frac{dpO_2}{dr} > 0$. Finally, to include capillary bed data points, we added all $pO_2$ measurements within the low 35% tail of each $pO_2$ map. Taken together, these steps resulted in exclusion of data points around additional oxygen sources within the capillary bed deviant from the model assumptions. Included $pO_2$ data were then collapsed to generate a radial gradient for estimation of $CMRO_2$ with the ODACITI model for each periarteriolar grid. When multiple acquisitions of the same plane were available, $CMRO_2$ estimates were averaged.

## Synthetic data

To validate the estimation of $CMRO_2$ using ODACITI, we generated synthetic $pO_2$ data by solving the Poisson equation for a given (space-invariant) value of $CMRO_2$ and chosen geometry of vascular sources and measurement points. This was done numerically using the finite element software package FEniCS [23] as described in our recent publication [24]. Briefly, we solve the variational formulation of the Poisson equation where $CMRO_2$ is constant, intravascular $pO_2$ is fixed, and there is no pressure gradient at the vessel wall boundary. Then, if $V$ is a space of test functions $[v_1, \ldots v_N]$ on the computational domain $\Omega$, we can find $pO_2$ such that $\int_\Omega \nabla pO_2 * \nabla v_i$ is equal to $CMRO_2$ scaled by a number that depends on the test function $v_i$. FEniCS provides the solution on an unstructured finite element mesh. Experimental data, in contrast, are measured on a Cartesian grid. Therefore, we transferred the synthetic data generated by FEniCS to a Cartesian grid similar to that used in our experiments. Additive Gaussian noise was implemented using the *normrnd* MATLAB function. For each synthetic data point in the grid, we drew a random number from a Gaussian distribution with the $pO_2$ value at this

point as the mean (μ) and a standard deviation of σ = 2. Afterwards, we replaced the $pO_2$ value by (μ + σ).

## Immunohistochemistry

COX labeling was performed in 4 C57BL/6J mice age-matched to the mice used for in vivo imaging. Mice were sacrificed and transcardially perfused with phosphate-buffered saline (PBS) containing 0.4% heparin and 2% sucrose, followed by 2% sucrose and 4% formaldehyde in PBS. The brain was removed and post-fixed with 2% sucrose and 4% formaldehyde in PBS overnight at 4°C. The brain was then washed 3 times with PBS and left for at least 1 d at 4°C in PBS containing 20% sucrose. Afterwards, 50-μm coronal sections were cut with a vibratome and transferred to PBS containing 0.5% bovine serum albumin (BSA; Sigma, A1470). Sections were permeabilized with 1% Triton X-100 (Sigma, T9284) and 0.5% BSA in PBS for 1 h. Antigen unmasking was performed with 10 mM Na-citrate buffer for 20 min at 75°C. After washing slices 3 times with PBS containing 0.3% Triton X-100 and 0.5% BSA (washing solution), slices were incubated in blocking solution (PBS with 10% normal donkey serum, 0.3% Triton X-100, and 0.5% BSA) overnight at room temperature. Thereafter, slices were incubated with rabbit polyclonal antibody against COX (Abcam ab16056, diluted 1:400 in blocking solution) for at least 24 h. Subsequently, sections were washed 3 times in washing solution for 10 min and incubated with donkey anti-rabbit IgG-conjugated Alexa Fluor 594 (Thermo Fisher Scientific) diluted 1:400 in blocking solution for at least 24 h at room temperature. At the end, sections were incubated with Hoechst 33342 (Thermo Fisher Scientific) for 15 min to label nuclei and then washed 3 times in washing solution before being transferred to Superfrost microscope glass slides (Fisher Scientific) and mounted with ProLong Gold mounting medium (Thermo Fisher Scientific). Slices were imaged using 2-photon microscopy. Alexa Fluor was excited at 800 nm and imaged with an Olympus 20× water-immersion objective (XLUMPlanFLNXW, NA = 1.0). We collected stacks of 2-photon images stepping though the section at 2-μm steps. We used a resolution of 512 × 512 pixels to image an 813 × 813 μm field of view; 2–4 averages were acquired at each imaging plane.

To quantify COX labeling, we analyzed 18 sections from 4 mice. We first masked cell nuclei that appeared dark in COX images. The masks were obtained by thresholding Hoechst images. In addition, large blood vessels were removed by manual segmentation. Fluorescence intensity profiles as a function of the cortical depth were obtained by drawing a rectangular ROI aligned with the cortical surface and computing averaged intensity along each row within the rectangle.

## Statistics

Statistical analysis was performed in R (https://www.r-project.org/) using a linear mixed effects model implemented in the lme4 package, where trends observed within a subject (e.g., the dependence of $CMRO_2$ on depth) were treated as fixed effects, while the variability between subjects was considered as a random effect. Observations within an animal subject were considered dependent; observations between subjects, independent.

## Supporting information

**S1 Appendix. Derivation of the ODACITI model.**
(DOCX)

**S1 Data. Numerical values for Figs 1G–1I, 2B, 2C, 2E, 2F, 3B–3F, S3, S4, S5C, S5D, S6A–S6C, S7C, S7D, S8B–S8E, S9C–S9E and S10 are provided as individual data sheets within**

**the same file.**
(XLSX)

**S1 Fig. Optical window with a silicone access port.** (A) Top view of the window before filling the injection port with silicone. There are three 3-mm glass coverslips stacked together and glued to a 5-mm glass coverslip. The 3-mm stack is beveled to guide the pipette at an angle through the port. (B) A side view of the window; the port is filled with silicone. (C) A top view of an implanted window; surface blood vessels are visible under the glass. (D) An image of surface vasculature within the same window calculated as a maximum intensity projection (MIP) of a 2-photon image stack 0–300 μm in depth using a 5× objective. Individual images were acquired every 10 μm. Fluorescence is due to intravascular FITC. (E) Consecutive images obtained 43 d apart demonstrate the stability of surface vasculature over time.
(TIF)

**S2 Fig. Example dataset for 1 arteriole across depths.** (A) Intravascular FITC images for 6 imaging planes (cortical surface, 100 μm, 200 μm, 300 μm, 350 μm, 400 μm). Arrows point to a diving arteriole that branches into 2 between 100 and 200 μm. (B) Phosphorescence images for each of these planes. (C) SR101 images; for 2 of the planes SR101 images were not acquired. (D) A square measurement grid of $20 \times 20$ points for each of the imaging planes. Photon counts are superimposed on corresponding vascular FITC images. (E) Calculated $pO_2$ values superimposed on corresponding vascular FITC images.
(TIF)

**S3 Fig. Average cortical depth profile of $pO_2$.** Mean $pO_2$ calculated by averaging all points measured at the same cortical depth, plotted as a function of depth; each line corresponds to a set of measurements acquired along 1 diving arteriole. Subjects are color-coded; in 3 cases, 2 arterioles were measured per subject. Note that these values do not represent the mean tissue $pO_2$ because they are biased towards highly oxygenated periarteriolar regions (all measurement grids in this study were centered around penetrating arterioles). Numerical values are provided in S1 Data (sheet S3).
(TIF)

**S4 Fig. The effect of $R_t$ on CMRO$_2$ estimates.** (A) Synthetic $pO_2$ data based on the Krogh–Erlang model (CMRO$_2$ = 2 μmol cm$^{-3}$ min$^{-1}$, $R_t$ = 80 μm, $P_{ves}$ = 60 mmHg, $R_{ves}$ = 10 μm) are plotted against the distance from the arteriole. A constant $pO_2$ is assumed for $r > R_t$. (B) The estimated CMRO$_2$ after applying ODACITI (green) and the Krogh–Erlang model (black) to the synthetic data in (A) as a function of different assumed $R_t$ values (the true $R_t$ = 80 μm). Numerical values for (A) and (B) are provided in S1 Data (sheets S4A and S4B).
(TIF)

**S5 Fig. $pO_2$ measurements at the cortical surface.** (A) Intravascular FITC image of a small surface arteriole (left) and a corresponding phosphorescence image (right). (B) A square measurement grid overlaid on the FITC image from (A). Left: Photon counts. Right: Calculated $pO_2$ values. Scale bar = 50 μm. (C) Histogram of $pO_2$ values corresponding to (B). (D) Overlaid histograms from each surface measurement plane (corresponding to individual arterioles, gray) and superimposed average (red). Numerical values for (C) and (D) are provided in S1 Data (sheets S5C and S5D).
(TIF)

**S6 Fig. Cumulative radial $pO_2$ profiles for each layer.** (A) All data points included in the CMRO$_2$ estimation in Fig 3; the mean (calculated using 2.5-μm binning) is superimposed in thick black. (B) The same as (A) for layer II/III. (C) The same as (A) for layer IV. Numerical

values for (A–C) are provided in S1 Data (sheets S6A–S6C).
(TIF)

**S7 Fig. Quantification of laminar cytochrome oxidase (COX) density.** (A) From left to right: 50-μm thick cortical section immunolabeled for COX; cell nuclei (Hoechst); overlaid COX (green) and Hoechst (blue); thresholded Hoechst image; masked COX image used for quantification. Scale bar is 100 μm. (B) For COX quantification, we defined a rectangular ROI (pink) aligned with the cortical surface (left). Then, we sliced the ROI in 1-μm slices in the horizontal (laminar) direction and averaged the mean intensity within each slice (outside the Hoechst mask), resulting in the laminar intensity profile shown on the right. Scale bar is 100 μm. (C) Overlaid laminar intensity profiles for 18 ROIs (color-coded) from 4 brains. The mean is shown in thick black. (D) Boxplot of data shown in (C) sorted by cortical layer as indicated in (C). Numerical values for (C) and (D) are provided in S1 Data (sheets S7C and S7D).
(TIF)

**S8 Fig. Estimation of signal-to-background ratio (SBR).** (A) For each imaging plane, the 8 most delineated (dark) cells were manually selected from phosphorescence intensity reference images. A 50-μm line was drawn across each cell (cyan) to extract an intensity profile. (B) The intensity profiles across the 8 cells were averaged and normalized to the mean intensity of neuropil (10–20 μm from center). The cortical layers are color-coded, and the mean of each cortical layer is overlaid in a thick line. (C) Comparison of data acquired with an objective with NA = 1.0 (solid lines) and NA = 0.5 (dotted lines). (D) Quantification of SBR as the mean intensity within cells (0–5 μm from center) divided by the surrounding tissue intensity (10–20 μm from center). Each point represents 1 imaging plane. The data are color-coded by the objective NA. (E) The same data as in (D) sorted by cortical layer (mean + SD). Numerical values for (B–E) are provided in S1 Data (sheets S8B–S8E).
(TIF)

**S9 Fig. Accuracy of pO$_2$ estimation as a function of number of excitation cycles.** (A) Left: An imaging plane 100 μm below the surface. Fluorescence is due to intravascular FITC. Middle: A grid of measurement points overlaid on the FITC image. Right: Photon count overlaid on the FITC image. (B) pO$_2$ values overlaid on the FITC image from (A). (C) Phosphorescence decays for 2 points labeled in (B). Data (points) and fit (lines) are overlaid. Lower pO$_2$ corresponds to slower decay (blue). (D) Residual error over time for "point 1" from (C). (E) Residuals plotted against the number of excitation cycles. For each time bin, residuals were calculated as a sum of fractional differences: First, we computed the absolute value of the difference between the data and the single exponential fit. Next, we normalized this value by that of the fit. Finally, we summed the normalized fractional differences over the time bins. The plot shows an average across all 400 points in the measurement grid shown in (A) and (B). These results are in general agreement with quantification of the error in pO$_2$ estimation as a function of the number of excitation cycles, which was done in our prior study [18]. Numerical values for (C–E) are provided in S1 Data (sheets S9C–S9E).
(TIF)

**S10 Fig. Experimental measurement of R$_t$.** $R_t$ was estimated from vascular FITC images as the average radial distance from the center of the arteriole to the nearest capillary within the segmented ROIs that were used for CMRO$_2$ estimation. Data are color-coded by subject. Numerical values are provided in S1 Data (sheet S10).
(TIF)

## Acknowledgments

The authors acknowledge that some of the data analysis reported in this paper was performed on the Shared Computing Cluster, which is administered by Boston University's Research Computing Services.

## Author Contributions

**Conceptualization:** David A. Boas, Richard B. Buxton, Gaute T. Einevoll, Anders M. Dale, Sava Sakadžić, Anna Devor.

**Data curation:** Philipp Mächler, Natalie Fomin-Thunemann, Layth N. Amra, Emily A. Martin, Ichun Anderson Chen, Payam Saisan, Sava Sakadžić, Anna Devor.

**Formal analysis:** Philipp Mächler, Natalie Fomin-Thunemann, Layth N. Amra.

**Funding acquisition:** Anna Devor.

**Investigation:** Philipp Mächler, Natalie Fomin-Thunemann, Ikbal Şencan-Eğilmez.

**Methodology:** Natalie Fomin-Thunemann, Martin Thunemann, Michèle Desjardins, Kıvılcım Kılıç, Ikbal Şencan-Eğilmez, Baoqiang Li, Qun Cheng, Kimberly L. Weldy.

**Project administration:** Anna Devor.

**Resources:** Martin Thunemann, Ichun Anderson Chen, Payam Saisan, John X. Jiang.

**Software:** Martin Thunemann, Marte Julie Sætra, Michèle Desjardins, Baoqiang Li, Payam Saisan, Anders M. Dale, Sava Sakadžić.

**Supervision:** Kıvılcım Kılıç, Sava Sakadžić, Anna Devor.

**Visualization:** Layth N. Amra.

**Writing – original draft:** Philipp Mächler, Natalie Fomin-Thunemann, Layth N. Amra, Emily A. Martin, Ichun Anderson Chen, John X. Jiang, Anna Devor.

**Writing – review & editing:** Philipp Mächler, Natalie Fomin-Thunemann, Martin Thunemann, Michèle Desjardins, Kıvılcım Kılıç, Layth N. Amra, Emily A. Martin, Ichun Anderson Chen, Ikbal Şencan-Eğilmez, Baoqiang Li, Payam Saisan, John X. Jiang, Qun Cheng, Kimberly L. Weldy, David A. Boas, Richard B. Buxton, Gaute T. Einevoll, Anders M. Dale, Sava Sakadžić, Anna Devor.

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
