## [Editor Report · Decision Letter 0]

8 Oct 2021

Dear Dr Devor, 

Thank you for submitting your manuscript entitled "Microscopic Quantification of Oxygen Consumption across Cortical Layers" for consideration as a Short Report by PLOS Biology.

I have now had the chance to discuss your manuscript with other members of the PLOS Biology editorial staff as well as by an academic editor with relevant expertise and I am writing to let you know that we would like to send your submission out for external peer review.

Once your full submission is complete, your paper will undergo a series of checks in preparation for peer review. Once your manuscript has passed the checks it will be sent out for review. 

If your manuscript has been previously reviewed at another journal, PLOS Biology is willing to work with those reviews in order to avoid re-starting the process. Submission of the previous reviews is entirely optional and our ability to use them effectively will depend on the willingness of the previous journal to confirm the content of the reports and share the reviewer identities. Please note that we reserve the right to invite additional reviewers if we consider that additional/independent reviewers are needed, although we aim to avoid this as far as possible. In our experience, working with previous reviews does save time. 

If you would like to send your previous reviewer reports to us, please specify this in the cover letter, mentioning the name of the previous journal and the manuscript ID the study was given, and include a point-by-point response to reviewers that details how you have or plan to address the reviewers' concerns. Please contact me at the email that can be found below my signature if you have questions. 

Please re-submit your manuscript within two working days, i.e. by Oct 12 2021 11:59PM.

Kind regards,

Lucas

Lucas Smith

Associate Editor

PLOS Biology

lsmith@plos.org

---

## [Decision Letter · Decision Letter 1]

7 Dec 2021

Dear Anna,

Thank you for submitting your manuscript "Microscopic Quantification of Oxygen Consumption across Cortical Layers" for consideration as a Short Report at PLOS Biology. I apologize again for our delay in sending you a decision - I had a hectic week last week trying to catch up after Thanksgiving, and wasn't quite able to finalize our discussion until today. Your manuscript has been evaluated by the PLOS Biology editors, an Academic Editor with relevant expertise, and by several independent reviewers. 

The reviews are appended below. As you will see, the reviewers have commented that the topic examined here is important and that the study is generally well done. However, they have also raised a number of important concerns which would need to be thoroughly addressed before we can consider the manuscript for publication. The reviewers note issues with the model and simulations, high variability in the data related to a small sample size, and limitations with the contrast during deep measures which collectively undermine the strength of the conclusions. Moreover, Reviewer 1 has commented that you should compare the CMRO2 against other measures, such as mitochondrial density.

As a note, after discussion with the Academic Editor, we do not think that you need to perform measures during stimulation, as suggested by Reviewer 1 and discussed by Reviewer 3. However, without testing CMRO2 under stimulation, we agree with Reviewer 1 that the relevance of your findings to fMRI is less clear. Therefore, the motivation for the study should be changed to de-emphasize the link to fMRI and to instead highlight the implications of this work to advancing understanding of cortical physiology/developing the experimental and analytical models used here.

In light of the reviews, we will not be able to accept the current version of the manuscript, but we would welcome re-submission of a much-revised version that takes into account the and thoroughly addresses the reviewers' comments. We would expect a revised manuscript to add more mice, to compare the measures of CMRO2 against other measures such as mitochondrial density, to clarify issues around the model, and to shift the motivation/intro/discussion. We cannot make any decision about publication until we have seen the revised manuscript and your response to reviewers. Your revised manuscript is also likely to be sent for further evaluation by the reviewers.

We expect to receive your revised manuscript within 3 months. 

**IMPORTANT - SUBMITTING YOUR REVISION**

*Re-submission Checklist*

*Published Peer Review*

*PLOS Data Policy*

*Blot and Gel Data Policy*

Sincerely,

Lucas Smith

Associate Editor

PLOS Biology

lsmith@plos.org

REVIEWS:

Reviewer #1: Machler et al. made quantitative measurements of oxygen concentration around a penetrating artery in multiple layers of brain tissue. They use the results to fit a model of oxygen delivery/consumption that takes into account the role O2 delivered by the capillaries distant from the artery, and find that the baseline CMRO2 is lower in the deep layers. This is motivated by the need to understand laminar fMRI signals, and where it is important to know CMRO2 on a layer-by-layer basis. the model is simple yet plausible. The data is nice, but very minimal. The experiments lack functional activation, as CMRO2 is known to change and the laminar-level differences might change drastically from rest. Without these key experiments, the impact of this manuscript is substantially lower. 

Major issues

The authors motivate their study by saying that the quantitative measurements of O2 levels - combined with models of O2 consumption in the tissue - are key input into understanding laminar fMRI. I don't think the results are meaningful for laminar fMRI for several reasons. First, I am not sure the region that the authors simulate will even contribute to the BOLD signal. The hemoglobin is saturated in the arteries, and the region they are simulating is largely devoid of capillaries and totally devoid of veins, whose higher deoxyhemoglobin concentrations will generate contrast. So BOLD fMRI is effectively blind to this region. Second, when using laminar fMRI in the context of a task, both the resting and the task-evoked signal will depend on CMRO2, and CMRO2 will change with the task, potentially in layer-by layer fashion. Measuring CMR02 at rest is only half of the metabolism story. While we know a fair bit about differences in CBF and vascular density across layers, one needs to know how CMRO2 changes with stimulation in order to say anything about the basis of laminar BOLD signals. Without some comparisons to a sensory-evoked condition, I don't see these measurements as being useful to the laminar fMRI community.

The model and the data fitting assumes that CMRO2 is constant at all distances from the arteriole. I don't think this likely, and the authors should consider the alternative hypothesis. One might expect that more metabolically demanding cell types (interneurons) to be clustered around the arteriole to take advantage of the higher O2 concentrations there. The authors could have relaxed this constraint and estimated how the observed O2 gradients might be impacted by CMRO2 variation with distance.

There are no quantifications or direct comparisons to previously published studies of laminar mitochondrial density, which could be a key indicator of actual CMRO2, and which would be sensible thing to test the model against. To start, one might expect that CMRO2 to linearly follow mitochondrial density. The Weber (2008 paper) is for primate visual cortex and the Land/Simons (1985) paper is not quantitative. 

Minor

Data will be shared upon publication- this does not conform to pls biology guidelines, needs more detail about the sharing procedure. The code for the model should be shared as well.

P 20 "silicon port" I think this should be silicone?

Reviewer #2: Summary

In their manuscript titled: "Microscopic Quantification of Oxygen Consumption across Cortical Layers", Mächler, Fomin-Thunemann present measurements of CMRO2 from the Layer I to the layer IV of the somatosensory cortex S1 of awake but unstimulated mice. They combined Two-photon Phosphorescence lifetime imaging with a recent, red-shifted oxygen sensor (Oxyphor 2P) and an innovative modeling approach to extract baseline consumption levels from oxygen consumption gradients present around pial arterioles. They report increased baseline oxygen levels in the layer IV (as compared to more superficial layers) as well as a reduced baseline oxygen consumption level at rest. This result is in contrast with established and published higher energy demands in the layer IV from Cytochrome oxidase staining and capillary densities studies. They conclude that transient and rapid increase in oxygen consumption upon sensory stimulation must account for this discrepancy. 

The quantification of baseline cerebral oxygen demand is an important endeavor which will yield many benefits for understand brain energy budgets and its regulation by biological processes. I applaud the efforts of the author in tackling this important question by integrating complex and modern instrumentation with modeling. Neuro-vascular units are tightly integrated components and precise compartmentalized and appropriately parametrized models are needed to push the field forward. Cerebral CMRO2 is a key variable in those models but has been hard to measure in vivo. This study presents a practical solution to solve this problem and will certainly move our understanding forward. 

In addition, the manuscript is clearly written and easy to read. It also references the relevant literature. All figures and captions are properly captioned. I commend the author for this clear presentation. 

While I have no concerns about the importance of this work, my main comments are methodological to make sure that the measure of CMRO2 was properly validated so that future work can rely upon it. 

MAJOR COMMENTS

1. Impact of baseline oxygen supply from nearby capillaries. 

The authors introduced a modification of the standard Krogh-Erlang model of O2 diffusion to account for the contribution of the capillary bed. As I understood it, in the ODACITI model, a key assumption is that for r<Rt, all oxygen molecules are provided by the pial arteriole and for r>Rt, all oxygen are provided by the capillary bed. As a result, oxygen flux at the boundary r=Rt is null. This hypothesis confused me somehow. In real tissue, shouldn't there be a transitional region where both capillary beds and the arteriole contributes? It seems to me like the capillary beds would need to have quite a complex geometry to satisfy this strict boundary condition. Presumably contribution from arteriole to CMRO2 gradually decrease with distance and capillary bed gradually takes over. 

This assumption is important because for the measured CMRO2 to be accurate, the measured gradient from the pial arteriole needs to account for 100% of the baseline consumption. If for example, nearby capillary beds would contribute to this baseline consumption, that will cause an underestimate of CMRO2, thereby impacting the reported decrease in oxygen consumption in the layer 4. 

The authors performed some simulation to try to test this caveat but I felt the simulation was a little limited. When does this assumption break? The author mentioned some parameters of the simulation matched real dataset (Page 10. "which is a typical size of the region around diving arterioles void of capillaries in mouse cerebral cortex.") but I was not sure whether this was a qualitative match or if some measurement of typical capillary density was used. 

This hypothesis also confused my interpretation and intuition of the equations. 

Page 24: "Third, we assume that for r>Rt the rate of O2 delivery by the capillary bed is uniformly distributed and equal to the rate of tissue O2 consumption."

Page 25: equation 8, why does pO2 depends on the radius for r>Rt if there is a perfect equilibrium between supply and demand for r>Rt. 

If the authors could provide more simulated controls and clarify the impact of these hypothesis on the interpretation of the data and associated equation, that will strengthen the use of this methodology in the future. 

MINOR COMMENTS

2. Page 6: "within ~40 min after the injection". What was the rationale for waiting such a long time before doing measurement? Does the injection impacts local oxygen levels? 

3. In the discussion, the importance of this absolute CMRO2 measurement for the fMRI BOLD signal is discussed. While this is a common statement in the neurovascular literature, I see more values in understanding fundamental neurovascular processes. BOLD signals are likely a mixture of tissue and blood oxygen levels and it is unclear to me how much it relates to absolute vs relative levels in oxygen consumption. Perhaps the authors could discuss in what ways they think this work will help understand BOLD more precisely.

4. Figure 1 is missing a few scale bars for a subset of panels. The depth used for each planar recordings on the figure is also missing. 

5. Figure 3. I was confused with panel B. How can the ODACITI fit predicts a small increase in pO2 if the surrounding tissue is in equilibrium with energy consumption? 

6. Figure 3. Are panel E and F duplicates? If that is the case, I felt duplicating that plot was confusing. 

7. Supplementary Figure 6. I applaud the quantification of residual measurement error. I was not entirely sure what were the units of those plots. Ideally, the error in decay time was converted to an averaged residual error in pO2 (using the calibration curves) and panel E is in mmHg. I was not sure given the figure and caption here. 

Reviewer #3: In the study by Machler et al., 2PLM measurements of PO2 are made across depth of S1 in awake mice in which the recent probe Oxyphor2P was injected directly into the cortex. They find that tissue PO2 immediately in the vicinity of the arteriole does not change significantly as a function of depth (consistent with previous vascular measurements), whereas "the low tail", shows an increasing PO2 concentration with increasing depth up to L4. Using a modified version of krogh model they develop termed "ODACITI", they estimate a decrease in CMRO2 from L1 to L4 at baseline. This is surprising as it contrasts with cytochrome oxidase staining which is highest in L4. 

Generally, the study is well executed and important for understanding O2 distribution in the cortex and for BOLD-fRMI interpretation. There are limitations to the study, many of which are addressed in the discussion. 

Comments:

A major limitation of the approach (as acknowledged by the authors) is that for the deeper measurements they begin to lose "contrast" due to excitation near the surface. This could potentially result in an artifact in which the signals appear to be smoothed out. How can we trust the measurements with this problem in mind?

It is unclear why they are using a 0.5NA objective (methods, p18), instead of a higher NA objective? This would greatly increase the out of focus excitation they are trying to eliminate for the spatial precision of their measurements.

It was not entirely clear to me if the 3D distribution of capillaries was taken into consideration in the model. As L4 has higher capillary density, could this explain the higher PO2 in the low tail of the distribution in L4?

The impact of the paper would be improved if they measured the tissue O2 change to a sensory stimulation. Although, this may be beyond the scope of the current study. 

Reviewer #4: This manuscript uses a model that was recently developed and published to quantify the oxygen metabolic rate in absolute units (u-mol/ml/min) in the top 4 layers of mouse cortex (L1, L2/3, L4) using a red-shifted exogenous phosphorescent oxygen sensor and two-photon fluorescence lifetime imaging. The authors report summary PO2 levels from near a feeding artery (upper 35% tail) and the tissue PO2 levels from tissue in these locations (lower 35% tail). These "raw" data show that tissue PO2 in L4 is highest and L1 is lowest despite arterial PO2 being roughly the same between these layers. As such, the model developed (based on the traditional and well established Krogh model) estimates that tissue PO2 in L4 is lowest, which is unexpected given the high amount of cytochrome-c present in L4 neurons. The work is fairly clear, the simulation helps setup and explain the model, before the awake head-fixed in vivo results are presented. Model selection is well justified (oxygen gradients around arteries provide direct information of tissue CMRO2). Potential weaknesses in the model selected are also reasonably explained. 

These findings are unexpected; however, the L1 CMRO2 estimates are considerably variable. It appears about 3 lines trend down in Fig 3C and 2 remain fairly flat. If this is indeed the case, this should be more clearly presented. The authors do use a reasonable statistical model to use all the data to test for significance. However, the results come from 4 (or 5) mice, a relatively low number, although much of their findings does support their claim. If possible, adding one more mouse would be helpful, and/or at least adding a sentence in the results (and may be discussion) to acknowledge L1 variability would be appropriate. Adding a panel in Figure 1 or in the Supplementary Material with the average PO2 vs. depth for each mouse (akin to Fig 3C) would be very valuable also. This is fantastic data from intricate and complex experiments.

It would be helpful to discuss how CMRO2 estimated from the neighborhood of arteries could be compared with tissue CMRO2 estimates obtained using the layer tissue PO2 measurements. For example, the authors should discuss how likely is the following rationale. If the venous PO2 is similar along any ascending vein and CBF is relatively uniform, how can these values be explained? Would this instead suggest that venous PO2 varies between layers? Venous PO2 could be estimated using a similar approach to the arterial gradient. Would heterogenous flow along capillary networks explain the difference in CMRO2 (the variability of tissue PO2 between layers does not appear that different -- error bar in figure 1). These scenarios might also help discuss the variability of the results presented.

Other comments are minor. Please use umol not um when placing units in the figures as this is easily confused with micro-meters. If you have estimates for the PSF dimension for imaging and FLIM, please provide them to understand the sampling FLIM volume vs. regular imaging.

---

## [Editor Report · Decision Letter 2]

12 Sep 2022

Dear Anna,

Thank you for your patience while we considered your revised manuscript "Microscopic Quantification of Oxygen Consumption across Cortical Layers" for publication as a Short Report at PLOS Biology. This revised version of your manuscript has been evaluated by the PLOS Biology editors and the Academic Editor. We are pleased to say that the Academic Editor is satisfied by the changes made in the revision and feels the manuscript is now ready for publication.

**However**, before we can accept your manuscript, we need you to address the following editorial requests in a revision that we think will not take very long. Please attend to the following editorial requests:

1) Title: After some discussion within the team, we feel the title should be edited to convey the key findings of your piece. If you agree, we suggest changing it to something like: "Baseline oxygen consumption decreases with cortical depth" or "Microscopic quantification reveals a decrease in basal oxygen consumption with cortical depth"

2) Ethics statement: In your ethics statement, please provide the approval number for the animal care and use protocol, approved by UCSD's IACUC. Please also include the specific national or international regulations/guidelines to which your animal care and use protocol adhered. Please note that institutional or accreditation organization guidelines (such as AAALAC) do not meet this requirement.

3) Data request: Thank you for depositing the data related to your study on BIL. Can you please provide us with a reviewer access token so that we can ensure that it is compliant with our data sharing policy?

We also ask that, at this stage, you please take a moment to read our Data policy, which requires that all data be made available without restriction, and confirm that these data meet our requirements: http://journals.plos.org/plosbiology/s/data-availability. For more information, please also see this editorial: http://dx.doi.org/10.1371/journal.pbio.1001797

a) Supplementary files (e.g., excel). Please ensure that all data files are uploaded as 'Supporting Information' and are invariably referred to (in the manuscript, figure legends, and the Description field when uploading your files) using the following format verbatim: S1 Data, S2 Data, etc. Multiple panels of a single or even several figures can be included as multiple sheets in one excel file that is saved using exactly the following convention: S1_Data.xlsx (using an underscore).

b) Deposition in a publicly available repository. Please also provide the accession code or a reviewer link so that we may view your data before publication. 

Fig 1G-I; Fig 2B-C,E-F; Fig 3B-F; Fig S3; Fig S4; Fig S5C,D; Fig S6A-C;Fig S7C-D; Fig S8 B-E; Fig S9C-E; Fig S10

>>Please also ensure that figure legends in your manuscript include information on where the underlying data can be found, and ensure your supplemental data file/s has a legend.

>>Please ensure that your Data Statement in the submission system accurately describes where your data can be found.

4) Data request: Please note that per journal policy, we do not allow the mention of "data not shown", "personal communication", "manuscript in preparation" or other references to data that is not publicly available or contained within this manuscript. Please either remove mention of these data or provide figures presenting the results and the data underlying the figure(s). 

I detected one instance where the manuscript refers to data not shown: "allowing to vary while fitting for CMRO2 did not reveal dependence of on cortical depth (not shown)."

We expect to receive your revised manuscript within two weeks. 

*Published Peer Review History*

*Press*

Sincerely,

Luke

Lucas Smith, Ph.D.

Associate Editor,

lsmith@plos.org,

PLOS Biology

---

## [Editor Report · Decision Letter 3]

30 Sep 2022

Dear Anna,

Thank you for the submission of your revised Short Report, "Baseline oxygen consumption decreases with cortical depth" for publication in PLOS Biology. Your revised manuscript has now been evaluated by the PLOS Biology editorial staff, and on behalf of my colleagues and the Academic Editor, Aniruddha Das, I am pleased to say that we can in principle accept your manuscript for publication, provided you address any remaining formatting and reporting issues. These will be detailed in an email you should receive within 2-3 business days from our colleagues in the journal operations team; no action is required from you until then. Please note that we will not be able to formally accept your manuscript and schedule it for publication until you have completed any requested changes.

As a final note, and as we discussed over email, to bring your manuscript into compliance with journal policy before accepting your study, I have made a minor edit to remove the reference to data not shown, on page 11 (deleting the sentence "allowing to vary while fitting for CMRO2 did not reveal dependence of on cortical depth (not shown)"). Please do let me know if you spot any issues with this change. 

PRESS

Sincerely, 

Lucas Smith, Ph.D., Ph.D.

Associate Editor

PLOS Biology

lsmith@plos.org